# Small Fish Big Impact: Improving Nutrition during Pregnancy and Lactation, and Empowerment for Marginalized Women

**DOI:** 10.3390/nu16121829

**Published:** 2024-06-11

**Authors:** Manika Saha, Heidi Ng, Emmanuel Nene Odjidja, Mallika Saha, Patrick Olivier, Tracy A. McCaffrey, Shakuntala Haraksingh Thilsted

**Affiliations:** 1Faculty of Information Technology, Monash University, Clayton 3168, Australia; 2Digital Nutrition Lab, Department of Nutrition, Dietetics and Food, Monash University, Notting Hill 3168, Australia; tracy.mccaffrey@monash.edu; 3Global Community Engagement and Resilience Fund (GCERF), 1202 Geneva, Switzerland; emmaodjidja@gmail.com; 4Department of Biology, Texas State University, San Marcos, TX 78666, USA; ldv51@txstate.edu; 5Action Lab, Faculty of Information Technology, Monash University, Clayton 3168, Australia; patrick.olivier@monash.edu; 6Nutrition, Health & Food Security Impact Platform, Consortium of International Agricultural Research Centers (CGIAR), Washington, DC 20005, USA; s.thilsted@cgiar.org

**Keywords:** first 1000 days of life, anemia, pregnant and lactating women, women empowerment, community-based intervention, food-based intervention

## Abstract

Undernutrition and micronutrient deficiencies such as anemia are considered significant public health challenges in Bangladesh, which enhancing fish consumption is a well-established food-based intervention to address these. This paper documents the establishment of community-based fish chutney production and reports the impact of its consumption on mid-upper arm circumference (MUAC) and hemoglobin (Hb) levels among targeted 150 pregnant and lactating women (PLW) in rural Bangladesh. A fish chutney was developed using locally available ingredients followed by a series of laboratory tests, including nutrient composition, shelf-life and food safety. A community-based fish chutney production process was designed to: (1) supply locally available ingredients for processing; (2) establish two fish drying sites; (3) initiate a community-based production site; and (4) distribute fish chutney to PLW for one year by six women nutrition field facilitators. Then a pre- and post-intervention study was designed for a selected 150 PLW to receive 30 g of fish chutney daily for 12 months. Differences in mean MUAC and Hb levels pre- and post-consumption were analyzed using one-way analysis of variance. Consumption of 30 g of fish-chutney resulted in significant increases of the mean values of Hb levels and MUAC among the targeted PLW.

## 1. Introduction

Malnutrition, both under- and over-nutrition, is one of the leading global health concerns, accounting for half of all child deaths (3.2 million globally), and 29% of women of reproductive age are affected by anemia [1]. If malnutrition remains unaddressed, the Food and Agriculture Organization (FAO) predicts that by 2030 the population of undernourished people will increase to 841 million from 678 million in 2018; however, this modeling has not accounted for the effects of the global COVID-19 pandemic, which is likely to lead to increases [2]. Maternal and child malnutrition during the first 1000 days of life, from the onset of pregnancy to the child’s second birthday and including the period of breastfeeding, has irreversible effects that include hampering fetal growth and optimal physical and cognitive development [3]. These adverse outcomes are inextricably linked to intergenerational, social, environmental factors and productivity growth [4]. Diverse and nutritious diets in the first 1000 days of life play a role in preventing all types of malnutrition [5,6]. Bangladesh, a low- and middle-income country, has made some visible progress in the economic and health sectors, although it has not kept pace in achieving significant improvements in reducing undernutrition [7]. The FAO definition describes this nexus of food and nutrition security which ‘exists when all people at all times have physical, social and economic access to food, which is safe and consumed in sufficient quantity and quality to meet their dietary needs and food preferences, and is supported by an environment of adequate sanitation, health services and care, allowing for a healthy and active life’ [8].

Maternal and child malnutrition, particularly micronutrient deficiencies such as iron and calcium, remain key public health problems in Bangladesh [9,10]. Low consumption of iron-rich, animal-source foods (ASF), such as fish and meat, are one major determinant of iron deficiency anemia in Bangladesh [11]. The increasing prevalence of anemia among pregnant women, from 39% in 2004 [12] to 49.6% in 2011 [13] as well as in half of all children (51%) [13], is a significant issue associated with maternal perinatal mortality, low birth weight (LBW) as well as severe impairment of child growth and brain development [1]. Therefore, ensuring maternal nutritional adequacy is important for both the mother and breastfed children.

A major underlying factor for micronutrient deficiencies in women and children in Bangladesh is a diet dominated by plant-based food sources, rice and a low intake of ASFs, with low dietary diversity and poor bioavailability of micronutrients [4]. A key determinant of undernutrition is the lack of access to sufficient, affordable and nutrient-rich foods, especially among marginalized populations such as rural women. In Bangladesh, low socioeconomic status, lack of knowledge of nutritious foods, availability, physical access and quality of the food consumed, as well as intra-household gender inequality, are major contributors to women’s poverty overall (in terms of poor health, nutrition outcomes and empowerment of women).

Community food-based nutrition interventions have the potential to yield positive impacts on the empowerment of women, micronutrient intakes, nutritional adequacy and socioeconomic statuses of marginalized women [14]. However, most community-based nutrition interventions are agricultural plant-based initiatives, such as promoting beta-carotene-rich sweet potato, which has resulted in improved micronutrient (vitamin A and iron) intakes among women in low- and middle-income countries [15,16]. For women enrolled in a Vulnerable Group Development (VGD) program in Bangladesh, there were significant improvements in food and nutrition security and prevalence of hunger following the consumption of zinc-fortified rice [17].

Globally, there is an increasing recognition of the critical contribution that fish, a highly nutritious and healthy ASF, makes to the world’s food systems to reduce micro- and macro-nutrient deficiencies. Fish is the most commonly consumed ASF in Bangladesh, with small indigenous fish species (SIS) making up about one-third of the fish consumption among low-income families or groups [18]. Usually eaten whole (with head, bones and organs), SIS, are a rich source of micronutrients such as iron, zinc, calcium, vitamin A, and vitamin B12, known as ‘problem’ nutrients [19]. The consumption of whole small fish was more nutritious than fish filets due to the removal of intestines and bones, which decreases the iron and calcium content [20]. Studies have shown that fish’s bioavailability of micronutrients, essential fatty acids and animal protein is significantly high [19,21]. For example, heme iron in some Bangladeshi SIS was almost double (93%) that of red meat (40–70%) [21]. The calcium concentration in small fish is higher than in milk, with fish and milk having similar high bioavailability [19].

In Cambodia, the consumption of fish-based products effectively treated acute malnutrition among young children [22]. During the wet and dry seasons, in many Southeast and South Asian countries, including Bangladesh, fish contribute to 50% or more of their main ASF [23]. However, it is crucial to consider the environmental factors, as Bangladesh is heavily influenced by flooding and monsoons, mainly affecting lower-income households, causing fluctuations in the supply and price of fish and the nutritional adequacy of people’s diet [4]. This seasonal supply of fish may be a major factor determining the ASF intake of both women and children.

In recent decades, the increase in the consumption of farmed fish in Bangladesh has resulted in a slight decrease in key micronutrient intake [24]. Furthermore, there is a lack of locally produced fish-based products from SIS that would improve both food and nutrition security. As Bangladesh is a fish-eating country, interventions focused on creating products from nutrient-dense SIS fish may be an effective avenue to combat malnutrition [6,7].

## 2. Study Objectives, Materials and Methods

The ‘Shiree Nutrition Innovation Project’ (called ‘Shiree Project’) is a community-based nutrition intervention (pilot project), within the Economic Empowerment of the Poorest Programme/Stimulating Household Improvements Resulting in Economic Empowerment (EEP/SHIREE) [25]. This initiative was directed toward improving the nutritional quality of the diets of pregnant and lactating women in rural Bangladesh by introducing an ASF product. The Shiree Project was led by WorldFish Bangladesh and partners. This project aimed to increase daily protein and micronutrient intake by providing a local, nutrient-rich fish-based food product among targeted pregnant and lactating women (PLW) during the first 1000 days of life (from the onset of pregnancy to the child’s second birthday and including the period of breastfeeding). The primary objective of our study was to evaluate the impact of a specific food product on the nutritional status of pregnant and lactating women, focusing on changes in Mid-Upper Arm Circumference (MUAC) and hemoglobin (Hb) levels. The benchmarks for success were predefined as a minimum increase of 1 cm in MUAC and 1 g/dL in Hb levels. These measures were chosen based on their clinical relevance and feasibility, and they informed the sample size and power calculations to ensure the study’s statistical robustness [26]. This paper outlines the development of a fish-based product for women; establishment of a community-based production unit; distribution of the food product, promotion of consumption of the product; and a field study on the outcomes of consumption of the fish-based product by a targeted group of PLW. This paper also presents lessons learnt and recommendations for future work. As part of the Economic Empowerment (EEP/SHIREE), this study went through all the ethical approval processes by the Research Review Committee and Ethical Review Committee of the institutional review committee board of the International Centre for Diarrhoeal Disease Research, Bangladesh (ICDDR,B) and the Bangladesh Medical Research Council (BMC). Informed consent was obtained from all participants involved in the study.

### 2.1. Study Site and Participants

This project was conducted in three upazilas (sub-districts) in Sunamganj district, north-eastern Bangladesh. Sunamganj is a large wetland area, with a rich source of freshwater fish species, including many SIS. Freshwater fish and dried fish from freshwater are culturally acceptable and popular in Sunamganj. Fish catching, selling and drying are major activities in Sunamganj. To obtain a heterogeneous sample of PLW, a two-stage cluster sampling was conducted across 35 villages (Figure 1).

The first stage identified an initial sample of 238 pregnant and 518 lactating women. In the second stage, inclusion and exclusion criteria were applied to select the PLW who would consume the fish chutney. The inclusion criteria were: women in the first trimester of pregnancy or lactating, up to 6 to 12 months postpartum. The exclusion criteria were: self-reported pregnancy and post-natal complications; and allergy or adverse reactions to the ingredients of the fish chutney. This information was collected using a pre-tested questionnaire. A screening questionnaire identified *n* = 179, of which 123 were pregnant and *n* = 56 lactating. During the study period, August 2014 to July 2015, 23 women dropped out, due to relocation or non-study-related reasons. At the end of the study, 152 women were in the lactational stage and four women who were in the lactational stage at the onset of the study were pregnant.

### 2.2. Stage 1: Fish Chutney Development

A ‘fish chutney’ for women was developed as a part of the Shiree Project. The fish chutney, based on a traditional ‘achar’ or pickle recipe, was developed by WorldFish Bangladesh using locally available ingredients: dried SIS, puti (*Puntius* spp.), onion, garlic, chili, oil and vinegar (Figure 2) [27]. Dried small fish provides a very concentrated source of the multiple nutrients found in fresh small fish. The chutney was developed to be eaten by PLW daily as a condiment to accompany the main meal.

Batches of fish chutney were analyzed for nutrient composition by AsureQuality Limited (New Zealand), the National Science Laboratory of Bangladesh and the Bangladesh Council of Scientific and Industrial Research (BCSIR) (Table 1). Tests were conducted for microbiological, chemical and pathogenic contamination as well as shelf-life stability by the ICDDR,B. Upon verification of food safety standards, acceptability trials were conducted among 20 PLW in the project area. Based on the responses, minor adjustments were made with respect to taste and flavor; texture and consistency; aroma and appearance to ensure high acceptability of the fish chutney among PLW.

### 2.3. Stage 2: Community-Based Fish-Chutney Production

Two community-based fish drying sites, each led by a local woman (not part of the study participants, *n* = 179) and her family, were established to ensure community engagement, capacity building, income-generating activities, women’s engagement, proper monitoring for quality control and supply chain management of the SIS and other ingredients; as well as sustainability of future dried SIS production (Table 2). Local fishermen, who normally sold fish to the local fish market, were engaged to supply the required fish to the community-based drying sites. A partnership with a community-based organization, Sunamganj Kallayan Staff Sangstha (SKSS) was established for production of the fish chutney (Figure 3). Facilities and procurement of equipment were established to ensure safe and hygienic fish chutney production. Training of the personnel and supervision of the fish drying, and production sites were regularly conducted by WorldFish staff.

An experienced local cook, assisted by four rural women (not part of the study participants, *n* = 179), prepared the fish chutney using dried fish and the other ingredients bought from a local market. After preparation, the fish chutney was stored in glass jars, each containing 280 g chutney, sealed with air-tight lids and appropriately labeled. The glass jars were packed in paper cartons and kept in a store room at room temperature, for up to one week before distribution. Standard operating procedures for all steps of the fish chutney production to delivery to women were set up with assistance of a food technologist, and adherence to these were put in place, through training, on-going guidance and monitoring.

The selected PLW (*n* = 179) received fish chutney for 12 months; a daily portion size of 30 g, equivalent to one heaped tablespoon per day. Each week, a jar of fish chutney (280 g) was distributed to each PLW by the field facilitator.

The 280 g fish chutney provided amounted to one week’s consumption and an extra 70 g, taking into account that in the Bangladeshi context, women often share some food with their children. The PLW ate the recommended portion size of fish chutney with their regular main meal, consisting mainly of the main staple food, boiled rice; and on a few occasions, chapati or white bread and some vegetables. A baseline study was conducted among the selected PLW (*n* = 179), in August 2014 and at end-line study (*n* = 156), in July 2015.

Six female nutrition field facilitators distributed the fish chutney jars to targeted PLW. These facilitators were trained to deliver key nutrition messages to the targeted PLW and the communities. In addition to the appropriate use of the fish chutney by PLW, key messages included: the importance of fish consumption in the first 1000 days of life, optimal care, and nutrition practices for PLW, hygiene and sanitation for food preparation and eating and the importance of exclusive breastfeeding. At the community level, cooking demonstrations of fish chutney preparation, tasting, guidance on serving size and ideas for incorporation with the main meal. These activities supported the enhancement of nutrition knowledge and behavior change in the communities.

### 2.4. Stage 3: Nutritional Outcomes Assessment

The six nutrition field facilitators regularly followed up the targeted PLWs by visiting their houses. The blood hemoglobin (Hb) concentration and mid-upper arm circumstance (MUAC) were measured at the before and end of the 12 months fish chutney consumption intervention. Blood Hb concentration (g/L) was obtained from a finger puncture blood sample, using the HemoCue hemoglobin photometer [32]. MUAC (mm) was measured using a numerical insertion tape (MUAC tape). MUAC is an easy measurement to take and a good indicator of muscle mass in the body, used for screening malnutrition in pregnant women. To measure the association between maternal MUAC and low birthweight, WHO recommended a cut-off MUAC value of <21 cm to <23 cm (<210 mm to <230 mm) for malnutrition in pregnant women, globally [33]. A study recommended a cut-off of <23 cm, for pregnant women in Africa and Asia [34]. The field facilitators were trained and supervised in the use of the MUAC tape and HemoCue machine.

### 2.5. Data Analysis

The results from the baseline and end-line surveys were analyzed using the Statistical Package for Social Sciences (SPSS version 20). The primary outcomes: blood Hb concentration and MUAC at baseline and end-line, blood Hb concentration and MUAC were computed along with corresponding standard deviations to provide the statistics of continuing indicators. The lower and upper bounds were reached using the explore function in SPSS at a 95% confidence interval. A one-way analysis of Variance (ANOVA) was calculated to understand the significance of variations that existed among the variables in the lower and upper bounds. A two-way comparative analysis was conducted to understand the difference between baseline and end-line data. Mean difference and percentage of difference were further calculated together with an independent *t*-test with the aim of determining the significance of the change between the baseline and the end-line. The level of significance was set at *p* < 0.05. To recommend the food product for sustained use, we determined that the minimal changes needed were: (i) an increase in MUAC by at least 1 cm and (ii) an increase in Hb levels by at least 1 g/dL [26]. Our sample size and power calculations were informed by these benchmarks. The calculation of power was informed by the prevalence of malnutrition using the prevalence of anemia among women in Bangladesh. With a prevalence of 42.4% [13], power analyses were conducted to ensure that the study was adequately powered to detect minimal changes in all outcomes used in this study. Specifically, we calculated the required sample size to achieve 80% power at a 5% significance level to detect the predefined changes in the levels of anemia using Hb levels.

## 3. Results

### 3.1. Description of Women Included in Study

The characteristics of women participating are summarized in Table 3. The mean age at baseline was 27.1 (±5.2). This changed at endline to 28.1 (±5.2) and this emanated from 28 women who had ended lactating at the time of the endline, therefore did not qualify for inclusion. No woman experienced adverse reaction from the fish chutney supplementation and no side effects were reported.

### 3.2. Effect of Fish Chutney on Mid-Upper Arm Circumference

At baseline, mean MUAC for all study PLW was 226.2 mm (95% CI: 169–268 mm), lower than the cut-off value of 230 mm. This increased by 4.7% (*p* > 0.001) to 237.2 at the endline.

Among pregnant women, MUAC was 226.9 mm (95% CI: 217.1–254.3 mm), higher than among lactating women; 209.4 mm (95% CI: 198.2–215.3 mm). Among pregnant women, MUAC increased to 238.5 mm (95% CI: 223.2–257 mm), and among lactating women, to 236.7 mm (95% CI: 218–253 mm) (Table 4). The mean increase for MUAC between baseline and end line was 5.11% among pregnant women and 13.1% among lactating women, which was statistically significant (*p* = 0.011) (Table 5).

### 3.3. Effect of Fish Chutney on Blood Haemoglobin Concentration

Overall, hemoglobin concentration levels were lower than acceptable threshold of 120 g/L. However, this increased significantly by 13.7% to 124.3 g/L (±13.7) (*p =* 0.003) at the end line (Table 6).

Among pregnant women, the mean Hb level was lower (107 g/L (95% CI: 96–136 g/L) than in lactating women 111.6 g/L (95% CI: 69–120 g/L) at baseline (Table 7). However, after 12 months of fish chutney consumption at the end line, the Hb blood concentration among both groups with the highest marginal increase seen in pregnant women (*p* = 0.034). While the percentage mean difference from baseline to endline among pregnant women was 16.6% lactating women was 11.2% which was not a statistically significant difference (*p* = 0.132).

## 4. Discussion

Our results indicate that the intervention of using fish chutney as a food-based micronutrient-rich food product could make a significant contribution to improving nutrition and micronutrient deficiencies, specifically iron deficiency anemia among PLW in Bangladesh. The predefined benchmarks of a 1 cm increase in MUAC and a 1 g/dL increase in Hb levels were met, indicating the program’s success. These results align with previous studies, underscoring the clinical significance of the observed changes [34,35]. The a-priori objectives and success measures provided a clear framework for evaluating the nutritional intervention’s impact and guiding future recommendations for sustained use. Following 12 months of fish chutney consumption, the prevalence of anemia among the PLW decreased significantly by about one third, from 72% to 28%. These findings contrast with other food-based intervention studies in South Asia, which show insignificant impacts on nutrition outcomes. For example, a study in India [36] that aimed to increase iron status among non-pregnant women by using food-based micronutrient snacks (e.g., iron-rich vegetables, fruit, and whole milk powder) found no association with Hb status. A similar intervention in rural Bangladesh, in which pregnant women were supplemented with iron-folic acid tablets and micronutrient powder, likewise did not show any significant change in Hb between supplemented and control women [37]. One explanation for our positive findings could be that the inclusion of fish in our food product enhanced the bioavailability of iron in fish itself and absorption of iron from other food sources consumed with fish [38]. Therefore, our intervention might have played a vital role in increasing the Hb level among the PLW. A study reported that the SIS used in our fish chutney, *Puntius* spp., has high levels of iron and calcium, which can meet the nutritional demands at critical periods in the life cycle. A daily serve of puti will be sufficient to fulfill 38% of a lactating woman’s iron needs and 86% of their calcium requirements [19]. Furthermore, in Bangladesh, the diet of the poorest is dominated by the staple food, rice (77–78% of total diet) and the intake of fish, the most common ASF among in Northern Bangladesh is low, at 12.8 g/day [18]. The production and provision of fish chutney, which contributes 24% of the recommended nutrient intake (RNI) of iron for PLW [27] per serving (30 g/day), may therefore be an accessible and effective solution to this low fish consumption. To our knowledge, there was no adverse reaction to the fish chutney in the PLW.

Our findings also suggest that fish chutney consumption among PLW may lead to MUAC improvements. According to our study design, it is difficult to determine explicitly whether the MUAC improvements were due to just the macro- (protein) or micro- (vitamin and minerals) nutrients or a combination of these nutrients together. A previous study in Bangladesh have shown that the provision of an orange-flavored micronutrient-fortified beverage, produced by Procter & Gamble in the form of a powder, to children and adolescent girls [39] resulted in an improvement in micronutrient deficiency anemia, measured using Hb and serum ferritin (for depleted iron stores), and increased MUAC. While a growing body of evidence has shown the efficacy of locally produced fish-based food products for the treatment and prevention of malnutrition for children, to our knowledge, this is the first locally produced fish-based food product piloted specifically for PLW. In our study, PLW reported higher acceptability of the product, attributing this to the spicy taste and flavor of fish chutney. We may assume that adding the fish chutney to the meal could have resulted in increased food intake, and perhaps contributing to increased MUAC in the PLW.

In addition to contribution to the RNI for iron, the fish chutney intake also contributed to the RNI of other micronutrients: 35% of the RNI for calcium, 21% for iodine, 12% for zinc, 10% for vitamin B12, 8% for polyunsaturated fatty acids (PUFAs) and 8% for protein, addressing the gaps in essential nutrients to ameliorate undernutrition and micronutrient deficiency (MND) among PLW. Intervention participants observed that infants born to the targeted women as ‘Pusti bacha’ (i.e., well-nourished children) by community members, as they looked much healthier compared to the other infants in the community, suggesting the perceived intergenerational impacts of the fish chutney on micronutrient outcomes.

Dried fish is not culturally accepted in many areas in Bangladesh and globally due its smell [40]. Further investigation on drying procedures can make it more acceptable in large populations around the world, and new methods using vacuum dryers may decrease the fishy smell from dried fish. Some SIS are highly rich in vitamin A, but sun drying destroys the Vitamin A content from the fish. A study recommends investigating the use of vacuum dryers to keep the high nutrient content and improve the flavor to make it more acceptable [41]. The 2015 Nutrition policy in Bangladesh especially focused on improving women and children’s nutrition [42], so the use of fish-based products and scaling up of nutrition-sensitive agriculture interventions could contribute to mitigating undernutrition and MNDs in the first 1000 days of life in Bangladesh. Funding is also an important determinant for scaling up the use of these fish-based products. Furthermore, this intervention can be disseminated by appropriate distribution channels, such as the World Food Programme, school feeding programmes, national food distribution programme, the US Agency for Progress programme, commercial sales, and into a therapeutic food used in clinical and community health services to improve moderate to severe acute malnutrition. Lastly, taking into account any necessary modifications of the recipe depending on the target groups.

One of the important outcomes of this intervention was the establishment of community-based fish-based products production and fish drying sites. Locally produced fish-based products can be an effective way to improve micronutrient deficiencies among PLW, warranting examination and adaptation in other Asian and African settings where SIS and small dried fish consumption are common. An African study found that fishing households tend to consume more fish, especially in poorer households [43]. However, an Indonesian study found that in local households, fish consumption is highly dependent on the success of the fishing activities of the men other than the availability and the money to purchase fish [23].

There are various contributors of fish consumption in low-income countries should also be considered, including affordability, accessibility and gender norms in local households when adapting this intervention at a larger scale [44]. A study in Bangladesh explored the socio-cultural practices and implications for equity in rural fish-farming households and found that the distribution of food is often dependent on the individual’s age and gender [45], affecting mostly PLW. There is a tendency to prioritize adult men, then younger children, over women [4,46], with men receiving larger portions of ASFs. In spite of that, dishes made from small fish, such as the fish chutney, would be much easily portioned among household members in comparison to large fishes or meat [47]. Women are broadly involved in aquaculture. However, they are often systematically disadvantaged in access to favorable livelihood opportunities across various segments of aquaculture value chains and undergo poor health and nutrition conditions [4]. Our study gives opportunities to the local rural women to get involved in different stages such as fish sorting, processing, cooking, packaging, and drying fish sites management. Indeed, women involved in the drying fish sites establishment reported to be more confident and now self-independent as they continue their business. They experienced an increased self-worth, which even affected the way people viewed them. Women involvement in agriculture and income generating activities in rural Bangladesh have shown similar positive outcomes in terms of increased women empowerment regarding improved income, community leadership and decision making over the production and at the household level [48,49]. Moreover, women involvement in dried fish value chain approaches and income generating activities are already common and playing a key role in women empowerment in Bangladesh. Women involvement in fisheries is a key to improving access. Currently, women engagement in the aquaculture sector is less than a quarter. The FAO reported that to improve the economic and nutritional needs of women at different ages [50]. Additional interventions targeting other vulnerable populations such as adolescent girls in schools and garment workers should be conducted in the future, as they are prone to undernutrition and MND such as anemia [40].

Equitable access to nutrient-rich foods is needed to survive and to thrive remains a global concern. Our findings suggest that fish-based products may be a way to tackle micronutrient deficiencies. There is great potential in fish-based product (e.g., fish chutney):A rich source of protein and essential micronutrients, while ingredients are locally available and easy to prepare, providing opportunity for dietary diversity and a micronutrient rich diet during the food scarcity period in dry seasons;Fish chutney has a long shelf life of about 5–6 months, it uses of mustard oil, turmeric, garlic and vinegar, which increases its shelf life, as well as reducing the workload of the women by storing;Ready-to-use: fish chutney can be added to any traditional meal, such as rice, bread or chapatis;Suitable for local small-scale fisheries and community-based production to expand to commercial production and larger markets;Opportunity for women participation at each stage of production, especially in dried fish value chain approaches, which are already common in Bangladesh [51] and establish women’s empowerment in the community.

The recommended portion size of fish chutney is 30 g. The production cost in a community-based small-scale production was approximately GBP £0.17 or $0.22 USD and larger scale-ups, such as through proper distribution channels, could reduce the production cost to a few pennies per serving. Studies show that appropriate investment on nutrition intervention over one’s life can show a return through increased lifetime economic productivity [52]. The FAO calculated that $1.2 billion USD a year globally investing on micronutrient supply can result in 13 times higher output ($15.3 USD) [50]. Every $1 USD investment in scaling up nutrition intervention for the group of women and children leads to an immediate return of $16 USD in benefits [48]. Unfortunately, this potential is often neglected in many current policies and discussions. However, in the 2021 policy brief by the Global Panel on Agriculture and Food Systems for Nutrition, the discussion is focused on opportunities to expand the aquaculture in low- and middle-income countries, with the potential to positively improve their food and nutritional security, economy, and environment [51].

Research has supported the potential of aquaculture to meet local demands for ASFs [7]. Studies found that small-scale fisheries increase the local supply of fish and provision of essential micronutrients to improve the nutritional adequacy of locals [53,54]. The long chain from supply to production could stimulate fish farmers to more SIS cultivation, which ultimately will play a crucial role in fisheries and maintaining the chain of biodiversity in the ecosystem. This may increase the SIS production in the household pond, dried fish site establishment and other stages of production, which can improve the community economy by creating income generating employment opportunities. Moreover, our local fish-based products’ intervention can be applicable and scaled up with appropriate context-based modifications in many countries in Asian and African settings where SIS and small dried fish consumption is common [55].

Similar studies based on small fish-based products were recently conducted in Cambodia and Kenya, such as the WinFood programme though the outcome was not as expected due to several constraints in implementation stages [55]. Our learnings from this intervention, there is a need for regular follow up to the PLWs by nutrition field facilitators, staff training, using HemoCue machine for measuring Hb level, strong monitoring, and coordination in different stages of the project implementation to improve the total project environment more favorable for the most positive outcomes. Moreover, the local establishment of community-based food production sites and the fish drying sites are the most significant contributions of the project, and it helps to maintain the supply and value chain management for raw ingredients and reduces the costs for production, transportation, and storage. One constraint is scarcity of SIS during the dry season, the availability is low, and the price increase is too high. Using industrial cold storage or setting up a large-scale cold storage facility can be a solution for storing large amounts of SIS for use during the dry season, although the electricity cost for storage will increase. Alternatively, large amounts of dried fish can be produced during peak fish season for use during the dry season.

It is inspiring that there are a substantial number of research interventions based on SIS products, such as the recently concluded WinFood programme in Cambodia and Kenya, although the outcome is not as expected due to several constraints in implementation stages [55]. These studies provide a good basis not only in this research area but also the direction of our future research. Moreover, there is ongoing household and community level training on small fish powder production currently under the NOURISH project in Cambodia [56], with partner branding organizations: Save the Children and WorldFish. Future studies should be conducted to see if there is any correlation of small fish consumption with breast milk quantity and overall child development. A randomized control trial needs to be arranged in future work with case and control groups to report absolute improvement of the outcomes. Furthermore, efforts to trial the distribution of these products at scale through the public sector, such as school feeding programs and national food distribution programs; or private sector, such as commercial production and sales, as well as clinical and community health services, are needed.

## 5. Conclusions

Attaining optimal nutrition is a powerful catalyst for reaching several of the Sustainable Development Goals (SDGs), especially SDG 2: “End hunger, achieve food security and improved nutrition and promote sustainable agriculture”, SDG 3: “Ensure healthy lives and promote well-being for all at all ages” and SDG 5: “Achieve gender equality and empower all women and girls” between now and 2030. Inclusion of a fish-based product on the plate of diverse, nutritious, and safe foods can contribute to combating undernutrition and micronutrient deficiencies. This paper presents the positive outcomes on improving nutritional status in PLW, particularly iron deficiency anemia. The importance of a food-based approach focusing on the first 1000 days of life has recently been recognized. Small fish-based products, such as fish chutney, is a rich source of multiple nutrients that are bioavailable and essential nutrients in the diet. This article documents the process, initiatives and lessons learnt from our community-based nutrition intervention of the development of a fish-based product at the ground level. This community level establishment includes the set-up of dry fish sites, community level production, involvement of community women in different stages of production, promotion, and distribution in a year-round, long-term and sustainable way.

This local fish-based product can bring a pathway to affordable and nutritious diets to combat the burden of malnutrition with additional scope to increase economic growth through income generating activities, as well as increasing women’s empowerment at the community level. Therefore, there is a need for a pathway to a sustainable and long term healthy, intellectual, and economically productive future for the individual, family, community, nations and towards a prosperous world, along with the global attention now needed to strengthen nutrition sensitive policies with commitment to increase funding to support research and act, in order to accomplish these goals.

## Figures and Tables

**Figure 1 nutrients-16-01829-f001:**
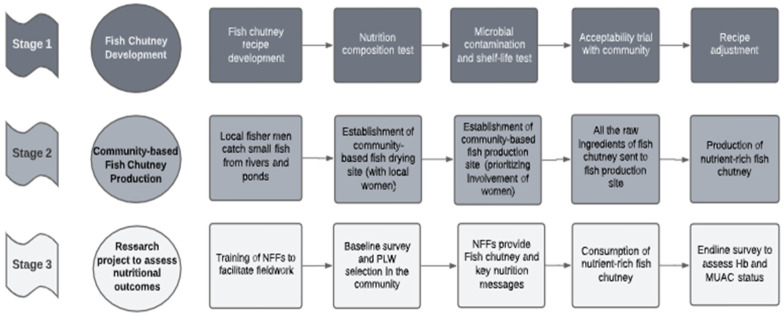
Fish Chutney Study Design Flowchart.

**Figure 2 nutrients-16-01829-f002:**
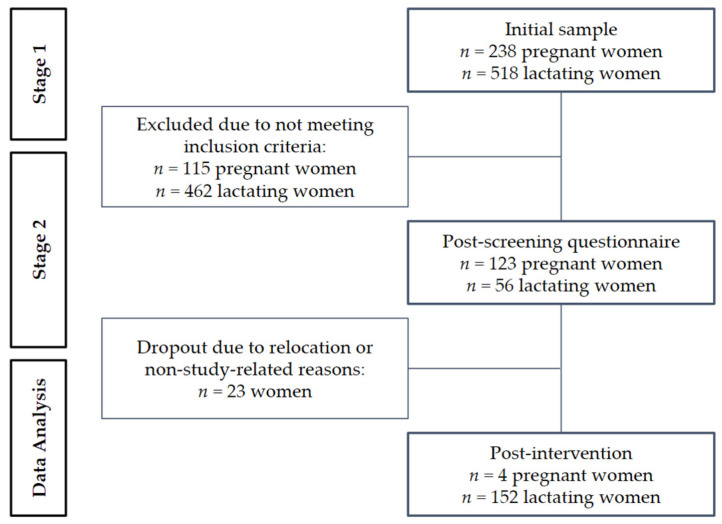
A flowchart of study recruitment.

**Figure 3 nutrients-16-01829-f003:**
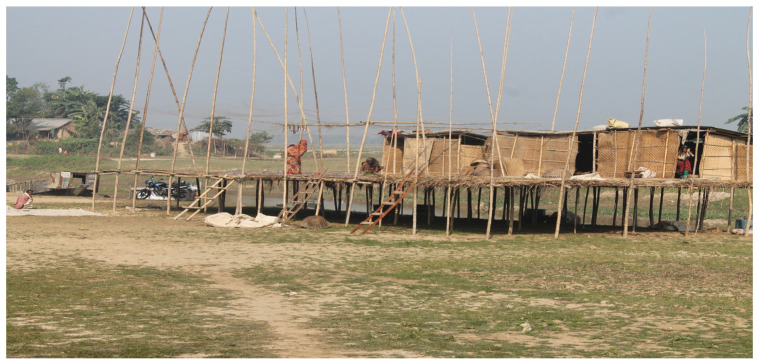
Fish Chutney Production Site. Photo credit: Taizul Islam (permission to use images was granted by participants).

**Table 1 nutrients-16-01829-t001:** Recipe of Fish-based Chutney and Complementary Food.

Ingredients	Composition by Weight, %	Composition per 30 g Serve, g
Fish Chutney		
Dried Jat puti fish Dried Jat puti fish	37	11.1 (44.4 g raw) ^a^
Onion, raw	37	11.1
Soyabean oil	15	4.5
Garlic, raw	7	2.1
Dried red chilli powder	4	1.2
Iodized salt	<1	<0.3

This table is a recipe adapted from Bogard et al. 2015 [27]. ^a^ Raw weight of fish is approximately 4 times the dried weight.

**Table 2 nutrients-16-01829-t002:** Nutrient Composition of Fish Chutney and Potential Contribution of a Daily Serve to Recommended Nutrient Intakes.

Nutrient	Unit	Composition per 100 g	Composition per 30 g Serve	Daily RNI	Contribution to RNI from 30 g Serve %
Macronutrients					
Energy	kJ	784	235	90948 ^a^	3
Protein	g	13.6	4.1	51.9 ^b^	8
Fat	g	10.3	3.1	68 ^c^	5
Micronutrients	mg				
Iron	mg	12	3.6	15 ^d^	24
Zinc	mg	3.1	0.9	7.9 ^c^	12
Calcium	mug	1200	360	1040	35
Iodine		140	42	200	21
Vitamin 812	mug	0.944	0.283	2.7	10
Fatty acids ^e^					
Total PUFA	g	5.8	1.7	22 ^f^	8
Total MUFA	g	2.6	1		
Total SFA	g	2.1	0.6	6144	23
C18:2n.6 (LA)	mg	4770	1431		
C20:4n-6 (AA)	mg	20	6		
18:3n-3 (ALA)	mg	420	126		
C20:5n-3 (EPA)	mg	nd	nd		
C22:6n-3 (DHA)	mg	17	5	200	3
Total n-6 PUFA	mg	4819	1446		
Total n-3 PUFA	mg	437	131	3072	4

Abbreviations: AA, arachidonic acid; ALA, α- linolenic acid; DHA, docosahexaenoic acid; EPA, eicosapentaenoic acid; LA, linoleic acid; MUFA, mono-unsaturated fatty acid; nd, not detected; PUFA, polyunsaturated fatty acid; RNI, recommended nutrient intake; SFA, saturated fatty acid. ^a^ Baseline energy requirements plus average energy requirements during pregnancy and the first 6 months of lactation, assuming a mean body weight of 45 kg (pre-pregnancy), a physical activity factor of 1.6 and age 18 to 29.9 years [28]. energy requirements > 6 months of lactation are highly variable depending on milk production, so were excluded. ^b^ Based on 0.83 g/kg/d, assuming a mean body weight of 45 kg (pre-pregnancy) plus 14.5 g/d throughout pregnancy and the first 12 months of lactation [29], ^c^ [30]. ^d^ Assuming 10% dietary bioavailability [31]. ^e^ Assuming moderate dietary bioavailability’ [31]. ^f^ Taken at the midpoint (9%) of the recommended range of PUFA as a percentage of total energy intake [30].

**Table 3 nutrients-16-01829-t003:** Descriptive information of study participants.

Women Characteristics	Baseline (B)	Endline (E)	Δ (B–E)
Pregnant (*n*) (%)	123 (68.7%)	4 (2.6%)	187.4%
Lactating (*n*) (%)	56 (31.3%)	151 (84.4%)	91.1%
Height (CM) ($\bar{x}$), SD (σ)	148.2 (±11.9)	150.3 (±6.5)	−2.2 (±14.2)
Total Participants (pregnant and lactating) (*n*)	179	155	14.4%

**Table 4 nutrients-16-01829-t004:** Baseline and Post-intervention comparative values for Mid-upper arm circumference in all study women (combined).

Variables	MUAC (mm) ^1^	$\mathrm{Mean}\;(\bar{x}$)	SD (σ)	% Mean Difference	*p*-Value ^2^ (Significance)
**All study women; pregnant and lactating**	Baseline (mm)	226.2	(±19.5)	4.9%	0.001
	Endline (mm)	237.2	(±25.2)		

^1^ MUAC: Mid-upper arm circumference; reference value 230 mm. ^2^ *p* < 0.05.

**Table 5 nutrients-16-01829-t005:** Baseline and Post-intervention comparative values for Mid-upper arm circumference in pregnant and lactating women in the study.

Variable		Baseline (*n* = 179)			Endline (*n* = 156)			% Mean Difference	*p*-Value ^2^ (Significance)
		$\mathrm{Mean}\;(\bar{x}$)	CI (95%)	SD (σ)	$\mathrm{Mean}\;(\bar{x}$)	CI (95%)	SD (σ)		
**Pregnant women**	MUAC ^1^ (mm)	226.9	217.1–254.3	±11.0	238.5	223.2–257.0	±15.9	5.11%	0.183
**Lactating women**		209.4	198.2–215.3	±15.1	236.6	218.9–253.0	±12.7	13.1%	0.011

^1^ MUAC: Mid-upper arm circumference; reference value 230 mm. ^2^ *p* < 0.05.

**Table 6 nutrients-16-01829-t006:** Baseline and Post-intervention comparative values for hemoglobin concentration in all study women (combined).

Variables		Hb (g/L)	$\mathrm{Mean}\;(\bar{x}$)	CI (95%)	SD (σ)	% Mean Difference	*p*-Value ^2^ (Significance)
**All study women; pregnant and lactating**	Hb ^1^ (g/L)	Baseline (mm)	109.5	83–128	±15.9	13.5%	0.003
		Endline (mm)	124.3	122–126	±13.7		

^1^ Hb: Hemoglobin; reference value 120 g/L. ^2^ *p* < 0.05.

**Table 7 nutrients-16-01829-t007:** Baseline and Post-intervention comparative values for hemoglobin concentration among pregnant and lactating women in the study.

Variable		Baseline (*n* = 179)			Endline (*n* = 156)			% Mean Difference	*p*-Value ^2^ (Significance)
		$\mathrm{Mean}\;(\bar{x}$)	CI (95%)	SD (σ)	$\mathrm{Mean}\;(\bar{x}$)	CI (95%)	SD (σ)		
**Pregnant Women**	Hb ^1^ (g/L)	107.0	96.2–136.1	±12.0	124.6	123.2–127.4	±12.7	16.6%	0.034
**Lactating women**		111.6	69.0–120.1	±16.2	124.1	123.3–125.1	±11.7	11.2%	0.132

^1^ Hb: Hemoglobin; reference value 120 g/L. ^2^
*p* < 0.05.

## Data Availability

The data used in this manuscript are not publicly available because of participants’ privacy concerns but are available on reasonable request.

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
