# Peer review of "Small Fish Big Impact: Improving Nutrition during Pregnancy and Lactation, and Empowerment for Marginalized Women"

_nutrients, 2024, doi:10.3390/nu16121829_

Round 1

Reviewer 1 Report

Comments and Suggestions for Authors

Thank you for the opportunity to review a MS entitled ‘Small fish big impact: Improving nutrition during pregnancy and lactation, and empowerment for marginalized women’ and I would like to recommend the MS for publication. This is a well written MS which provides uniquely good results on improvement in iron deficiency via simple, specifically processed fish food (chutney). The article is well written and meets the requirement of the public of Nutrients.

However, there are some minor remarks. It seems that the processed fish food is devised by the authors and is very practical, useful and can easily be provided to those who are starving in rural areas in Asia and Africa.

One sentence in the abstract requires a minimal language correction:

Line 28: by six women nutrition field facilitators

MUAC does not need to be shortened two times (lines 23 and 30).

Comment on statistics: the authors should provide what kind of t-tests did they perform (line 242).

The baseline demographic characteristics, obstetric history characteristics were omitted from the MS, which I recommend providing.

There are not presented pregnancy outcome characteristics on the pregnant and even on the lactating subpopulation which would be beneficial to know for the readers. At least birthweight of the neonates and gestational age at birth for the women are needed to present.

It should be provided information on what was the therapy length. How many weeks? In which gestational age and postpartum week were the subjects recruited?

Were there any other parameters followed up? Or only MUAC and Hb?

Figure 5- 7 are photos of everyday life of different persons which is not usual to publish in scientific journals, so I recommend excluding these from the final form of the article.

Author Response

We wish to thank the reviewers for their insightful comments and feedback on our work. It’s gratifying to have concrete ideas for improvement. We have summarised below the changes that have been made to the paper in response to the comments by reviewers. Wherever applicable, in our responses we refer the reader to the specific place in the paper where the changes or additions were made; in the paper itself, they are indicated in track changes.

Reviewers’ comment

Responses to the reviewers

R1

One sentence in the abstract requires a minimal language correction:

Line 28: by six women nutrition field facilitators

We have revised the sentence in line 28 of the abstract as suggested. The corrected sentence now reads: “A community-based fish chutney production process was designed to: 1) supply locally available ingredients for processing; 2) establish two fish drying sites; 3) initiate a community-based production site; and 4) distribute fish chutney to PLW for one year by six women nutrition field facilitators.”

R1

MUAC does not need to be shortened two times (lines 23 and 30).

Thank you for highlighting this redundancy. We have corrected the text by removing the second instance of the abbreviation. Now, MUAC is only abbreviated upon its first mention.

R1

Comment on statistics: the authors should provide what kind of t-tests did they perform (line 242). - Mean difference and percentage of difference were further calculated together with a T-test with the aim of determining the significance of the change between the baseline and the end-line. The level of significance was set at p<0.05.

We have clarified in the manuscript that an independent t-test was conducted to compare the two groups. Specifically, we calculated the mean difference and percentage of difference to determine the significance of the change between the baseline and end-line measurements. The level of significance was set at p<0.05. This information has been explicitly added to the relevant section of the manuscript under 2.5. 

R1

The baseline demographic characteristics, obstetric history characteristics were omitted from the MS, which I recommend providing.

There are not presented pregnancy outcome characteristics on the pregnant and even on the lactating subpopulation which would be beneficial to know for the readers. At least birthweight of the neonates and gestational age at birth for the women are needed to present.

We have updated the manuscript to include a comparative analysis of baseline and endline demographic characteristics among pregnant and lactating women. However, our study did not measure pregnancy outcomes such as birthweight and gestational age. We acknowledge that including these outcomes would have been beneficial and will consider incorporating such measures in future studies.

R1

It should be provided information on what was the therapy length. How many weeks? In which gestational age and postpartum week were the subjects recruited?

Thank you for your attention to these details. We have already included this information in the manuscript under sections 2.1, 2.3, and 2.4.

R1

Were there any other parameters followed up? Or only MUAC and Hb?

In our study, we focused exclusively on measuring and following up on MUAC and Hb levels. This information has been clarified and emphasised throughout the paper to ensure transparency and accuracy in reporting our study parameters. 

R1

Figure 5- 7 are photos of everyday life of different persons which is not usual to publish in scientific journals, so I recommend excluding these from the final form of the article.

In response to your suggestion, we have removed Figures 5-7 from the paper. We appreciate your attention to this matter and ensuring adherence to standard practices in scientific publishing.

We thank once again all the reviewers for their care, insights, and suggestions. We believe our revisions and responses clarify the central issues raised by the reviewers.

Reviewer 2 Report

Comments and Suggestions for Authors

An interesting study that reports data on a nutritional and educational strategy in pregnant and lactating women, which demonstrates how it affects Hb levels and arm circumference. Thank the authors for these data.

My main question is how intake, or the nutritional pattern, is controlled. Also, there are few variables that the authors can control that explain these data.

The introduction is well presented, however, the objectives of the study are not clear. Although, the paragraph of lines 112-124 could be included as objectives of the study. 

In the material and methods, it is not clear why the disparity in recruitment, pregnant and lactating women up to 1 year postpartum, there may be a great dispersion in nutritional requirements. It is not clear about the previous questionnaire mentioned in line 139, how it was applied, what evaluated its validity? I suggest that the authors report a flowchart of study recruitment.

Results: The main analysis of the data is not understood. For example, Table 1, the p-value compares the values ​​of the baseline with the post-intervention? Why are there then 2 p-values? The legend is not clear either. What does Hb 120 g/L mean? Is it the limit that the authors set? but then the means of the variable appear, why? the index "3" is <0.01, but they report the exact value of the P.

Minor comments:

- The size of the text in fig 1 must be increased.

- Figs 2 and 3 are really tables. It should be put as editable tables.

Comments on the Quality of English Language

In my experience, the English is understandable, with minor edits.

Author Response

We wish to thank the reviewers for their insightful comments and feedback on our work. It’s gratifying to have concrete ideas for improvement. We have summarised below the changes that have been made to the paper in response to the comments by reviewers. Wherever applicable, in our responses we refer the reader to the specific place in the paper where the changes or additions were made; in the paper itself, they are indicated in track changes.

Reviewers’ comment

Responses to the reviewers

R2

My main question is how intake, or the nutritional pattern, is controlled. 

This relevant information has been mentioned and clarified in lines 228-232 under section 2.3

R2

The introduction is well presented, however, the objectives of the study are not clear. Although, the paragraph of lines 112-124 could be included as objectives of the study. 

We have addressed this concern by clarifying the study objectives in the second section of the manuscript. We appreciate your attention to this aspect of the paper.

R2

In the material and methods, it is not clear why the disparity in recruitment, pregnant and lactating women up to 1 year postpartum, there may be a great dispersion in nutritional requirements. 

Our study focuses on the critical first 1000 days of life, from the onset of pregnancy to the child’s second birthday, which includes the period of breastfeeding. This information was initially mentioned in the introduction. However, as per the suggestion by Reviewer 2, we have now clarified this information in the material and methods section to provide a clearer understanding of our study population and rationale for recruitment disparity. We appreciate your attention to this detail.

R2

It is not clear about the previous questionnaire mentioned in line 139, how it was applied, what evaluated its validity? I suggest that the authors report a flowchart of study recruitment.

We have addressed this concern by adding a flowchart diagram of study recruitment to the manuscript, as suggested by Reviewer 2. The survey was conducted based on predefined inclusion and exclusion criteria to screen participant eligibility. It's important to note that no separate validation was conducted for the questionnaire used in this study. 

R2

Results: The main analysis of the data is not understood. For example, Table 1, the p-value compares the values ​​of the baseline with the post-intervention? Why are there then 2 p-values? 

Many thanks for this comment. The results have now been updated. In this version, we only compare outcome values between baseline and endline and determine the level of significance for pregnant women and those lactating. Furthermore, for better articulation, we have separated both outcomes. We appreciate your valuable input in improving the clarity of our results.

R2

The legend is not clear either. 

What does Hb 120 g/L mean? Is it the limit that the authors set? but then the means of the variable appear, why? the index "3" is <0.01, but they report the exact value of the P.

We have addressed these concerns by adding and editing details under tables 4 and 5 to improve clarity. Specifically, we have provided clearer explanations regarding the meaning of Hb 120 g/L and its relevance in the context of the study. Additionally, we have revised the presentation of means of the variable to ensure consistency and avoid confusion. Regarding the reporting of p-values, we have adjusted the legend to accurately reflect the significance level. We appreciate your valuable input in enhancing the clarity of our tables.

R2

Minor comments:

- The size of the text in fig 1 must be increased.

- Figs 2 and 3 are really tables. It should be put as editable tables.

  • We have addressed these comments by increasing the size of the text in Figure 1 (fish chutney flowchart) for improved readability.

  • Additionally, we have converted Figures 2 and 3 into editable tables to better align with the journal's formatting requirements. 

These changes ensure better clarity and compatibility with the journal's guidelines. We appreciate your attention to these details.

We thank once again all the reviewers for their care, insights, and suggestions. We believe our revisions and responses clarify the central issues raised by the reviewers.